# Electrochemically-stable ligands bridge the photoluminescence-electroluminescence gap of quantum dots

Chaodan Pu [1,3], Xingliang Dai[1,2,3], Yufei Shu[1,3], Meiyi Zhu[1], Yunzhou Deng [1,2], Yizheng Jin [1,2✉] & Xiaogang Peng [1✉]

Colloidal quantum dots are promising emitters for quantum-dot-based light-emitting-diodes. Though quantum dots have been synthesized with efficient, stable, and high colour-purity photoluminescence, inheriting their superior luminescent properties in light-emitting-diodes remains challenging. This is commonly attributed to unbalanced charge injection and/or interfacial exciton quenching in the devices. Here, a general but previously overlooked degradation channel in light-emitting-diodes, i.e., operando electrochemical reactions of surface ligands with injected charge carriers, is identified. We develop a strategy of applying electrochemically-inert ligands to quantum dots with excellent luminescent properties to bridge their photoluminescence-electroluminescence gap. This material-design principle is general for boosting electroluminescence efficiency and lifetime of the light-emitting-diodes, resulting in record-long operational lifetimes for both red-emitting light-emitting-diodes ($T_{95} > 3800$ h at 1000 cd m$^{-2}$) and blue-emitting light-emitting-diodes ($T_{50} > 10,000$ h at 100 cd m$^{-2}$). Our study provides a critical guideline for the quantum dots to be used in optoelectronic and electronic devices.

[1] Centre for Chemistry of High-Performance & Novel Materials, Department of Chemistry, Zhejiang University, Hangzhou 310027, China. [2] State Key Laboratory of Silicon Materials, Zhejiang University, Hangzhou 310027, China. [3] These authors contributed equally: Chaodan Pu, Xingliang Dai, Yufei Shu. ✉email: yizhengjin@zju.edu.cn; xpeng@zju.edu.cn

Colloidal quantum dots (QDs) are solution-processable semiconductor nanocrystals coated with a monolayer of surface ligands[1]. Progresses on synthetic chemistry of core/shell QDs have led to a unique class of emissive materials with efficient, stable, and high colour-purity photoluminescence properties[2–6]. Electroluminescence of the core/shell QDs is expected to harness the unique combination of their superior photoluminescence properties and excellent solution processability, enabling high-performance and cost-effective QD light-emitting-diodes (QLEDs), electrically driven single-photon sources, and potentially electrically pumped lasers[7–17]. However, design and fabrication of high-performance QLEDs remain to be challenging and empirical, which is considered as a result of unbalanced charge injection and interfacial exciton quenching[18]. Despite extensive efforts on material screening and device engineering, only a few exceptional examples are documented for red and green devices with decent external quantum efficiency and operational lifetime[9,10,15,17]. The operational lifetime of blue QLEDs remains to be very short (typically 100–1000 h for an initial brightness of 100 cd m$^{-2}$ to drop 50%)[10,19], shadowing the future of QLED technology. These facts indicate that, at the fundamental level, new design principles for QDs and/or QLEDs are necessary for bridging their photoluminescence-electroluminescence gap.

Photoluminescence and electroluminescence of QDs differ from each other in the excitation processes. In photoluminescence, an exciton is generated and confined mostly in the centre of a core/shell QD by absorbing a photon. In electroluminescence, an exciton is generated by injecting an electron and a hole separately into a core/shell QD. Surface ligands are inevitably accessible during the carrier-injection processes. This means that, though surface ligands are usually optimized for achieving high-performance photoluminescence, their electronic and electrochemical properties should be taken into consideration in terms of designing QDs for high-performance electroluminescence devices.

Here, we report that electrochemical stability of QDs under device operation conditions is an intrinsic but previously overlooked issue. Given inner inorganic lattice of a QD is electrochemically stable, the electrochemical degradation of QDs in the devices should be associated with their inorganic-ligand interface. We decide to systematically investigate the core/shell QDs with various combinations of outer shells and surface ligands, generalizing the concept of electrochemically inert ligands to be essential for bridging the photoluminescence-electroluminescence gap of QDs.

## Results
**Ligand-dependent electroluminescence of the CdSe/CdS QDs.** The recently developed CdSe/CdS core/shell QDs with nearly ideal photoluminescence properties[4] are applied as the first model system. The as-synthesized QDs, denoted as CdSe/CdS-Cd (RCOO)$_2$ QDs, comprise of 3-nm CdSe cores and 7-nm (10 monolayers) CdS shells coated with two types of carboxylate-based ligands. These two types of ligands are carboxylate ligands bonded onto the surface cadmium sites on the polar facets and cadmium-carboxylate ligands weakly adsorbed onto the non-polar facets[20,21] (Fig. 1a). An established surface-treatment procedure[22] converts the CdSe/CdS-Cd(RCOO)$_2$ QDs to the QDs solely with fatty amine ligands (CdSe/CdS-RNH$_2$) (Fig. 1a, and Supplementary Fig. 1). According to literature, only the standard Cd$^{2+}$/Cd$^0$ reduction potential of cadmium carboxylates is within the bandgap of CdSe and that of the other components of all ligands are far outside the bandgap (Supplementary Fig. 2)[23,24]. Optical measurements (Fig. 1b, c and Supplementary Fig. 3) show

that both types of the CdSe/CdS core/shell QDs, either in solution or in film, exhibit nearly identical, stable, and highly efficient photoluminescence. However, the CdSe/CdS-RNH$_2$ QDs and the CdSe/CdS-Cd(RCOO)$_2$ QDs deliver strikingly different electroluminescence performance in an identical QLED structure (Fig. 1d–g).

With the surface-treated QDs, namely CdSe/CdS-RNH$_2$ QDs, both current density and luminance of the QLEDs show a steep increase simultaneously at ~1.65 V (Fig. 1e). A device with the best efficiency shows a peak external quantum efficiency (EQE) of 20.2% (Fig. 1f), and the average peak EQE of our devices is 18.6%. Since the out-coupling efficiency of our devices is simulated to be ~23.5% by following a literature method[25], the average peak internal quantum efficiency (IQE) is estimated to be ~80%. The device lifetime ($T_{50}$, time for the luminance decreasing by 50%), is estimated to be ~90,000 h at 100 cd m$^{-2}$ by applying an acceleration factor of 1.8[16] (Fig. 1g).

The QLEDs using the as-synthesized CdSe/CdS-Cd(RCOO)$_2$ QDs deliver an extremely low peak EQE of 0.2% and a negligible $T_{50}$ of ~0.3 h at 100 cd m$^{-2}$ (Fig. 1f, g), indicating dominating and detrimental non-radiative recombination of the injected charges. It is interesting that the devices with the as-synthesized CdSe/CdS-Cd(RCOO)$_2$ QDs possess a voltage gap of 0.8 V between the threshold voltage for current raise and the turn-on voltage for light emission, while this voltage gap is almost zero for the QLEDs using the surface-treated CdSe/CdS-RNH$_2$ QDs (Fig. 1e).

**Operando electrochemical reaction of CdSe/CdS-Cd(RCOO)$_2$ QDs.** To explain the drastically different electroluminescence performance between the as-synthesized CdSe/CdS-Cd(RCOO)$_2$ QDs and the surface-treated CdSe/CdS-RNH$_2$ QDs (Fig. 1), we propose a scenario of electrochemical reduction of the cadmium-carboxylate ligands by the injected electrons. In literature, chemical reduction of cadmium-salt ligands on the cadmium chalcogenide QDs was reported[26,27]. As shown in Supplementary Fig. 2, the standard reduction potential of cadmium carboxylates (Cd$^{2+}$/Cd$^0$) is slightly below the bottom of the conduction band of bulk CdSe and CdS[23,26]. The proposed operando electrochemical reaction shall generate Cd$^0$ species, which effectively quench the luminescence of QDs[28], resulting in poor electroluminescence. A series of experiments are carried out to verify this hypothesis.

We monitor the relative changes of photoluminescence efficiency of the CdSe/CdS core/shell QDs in working QLEDs at different voltages[29] (experimental setup shown in Supplementary Fig. 4). The relative photoluminescence efficiency of the surface-treated CdSe/CdS-RNH$_2$ QDs is almost constant when the device bias is swept from 0 to 2 V. Above 2 V, the photoluminescence efficiency of the surface-treated CdSe/CdS-RNH$_2$ QDs show little yet reversible reduction, which is likely due to the increased electric field. In contrast, for the as-synthesized CdSe/CdS-Cd(RCOO)$_2$ QDs, the photoluminescence efficiency rapidly decreases when the bias is above 1.2 V (Fig. 2a), which equals to the threshold voltage associated with current raise and is much below the turn-on voltage for this type of QLEDs (Fig. 1e). The reduction of photoluminescence efficiency is not recovered when either zero bias (Fig. 2b) or a negative bias (data not shown) is applied to the QLED. Being irreversible at zero bias means that the possible redox products are stable without an external electric field, suggesting the excess holes are either remotely trapped in the hole-injection/hole-transport layers or resulting in stable oxidation products. The current of a diode is negligible under negative bias, which implies an attempt to reverse the suggested Cd$^{2+}$/Cd$^0$ reduction by applying negative bias would be difficult in the QLEDs.

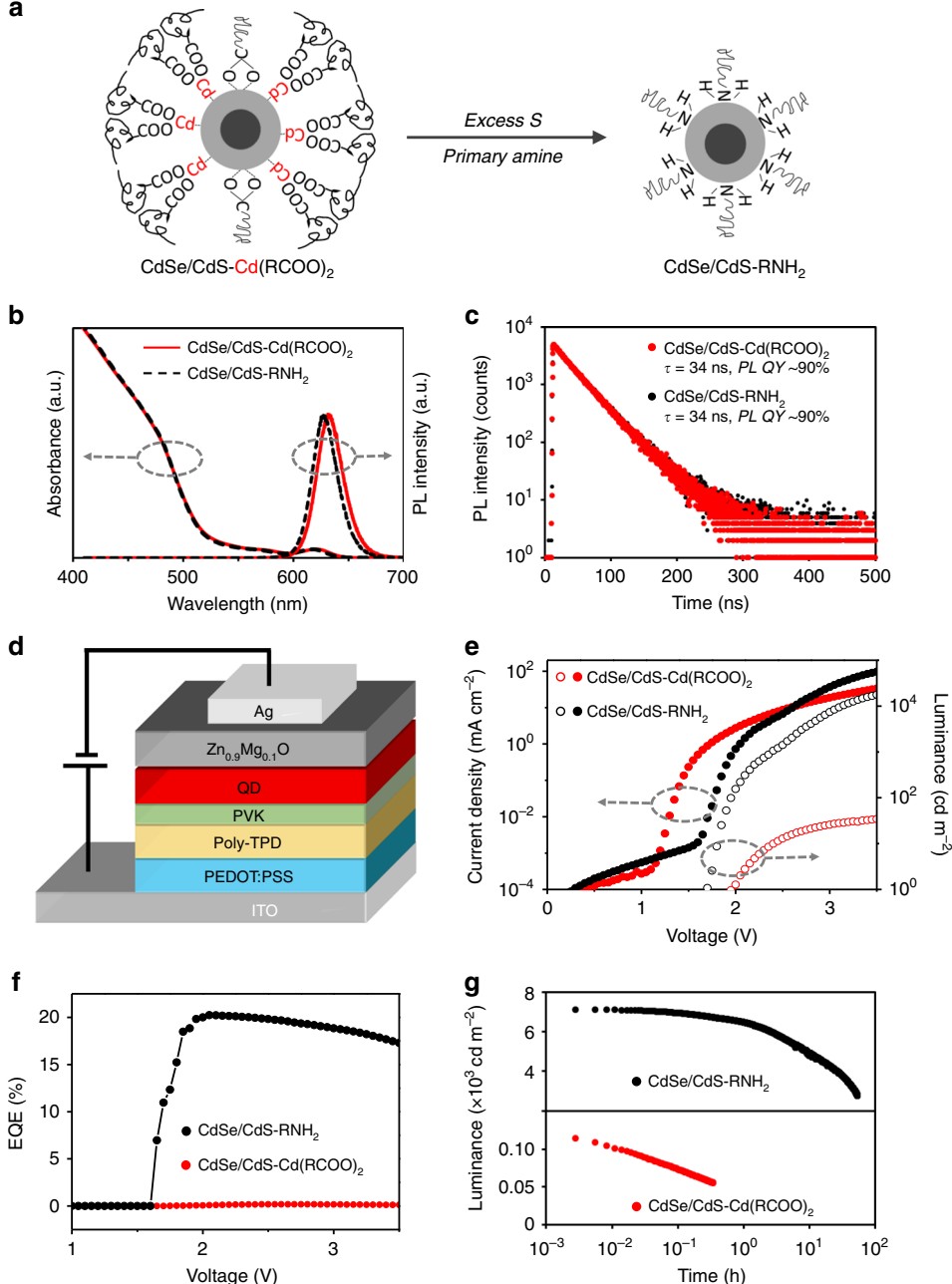

**Fig. 1 PL and EL properties of the CdSe/CdS-Cd(RCOO)$_2$ and the CdSe/CdS-RNH$_2$ QDs. a** Ligand exchange from cadmium-carboxylates (with a small amount of negatively charged carboxylates) to primary amines. **b** Absorption and steady-state photoluminescence (PL) spectra. **c** Time-resolved PL spectra with mono-exponential PL decay lifetime ($\tau$) and quantum yield (QY). **d** QLED structure: indium tin oxide (ITO)/ poly(ethylenedioxythiophene):polystyrene sulfonate (PEDOT:PSS, ~35 nm)/poly (N,N9-bis(4-butylphenyl)-N,N9-bis(phenyl)-benzidine) (poly-TPD, ~30 nm)/poly(9-vinylcarbazole) (PVK, ~5 nm)/ QDs (~40 nm)/Zn$_{0.9}$Mg$_{0.1}$O nanocrystals (~60 nm)/Ag. **e** Current density and luminance vs. driving voltage characteristics of the QLEDs. **f** EQE vs. driving voltage characteristics of the QLEDs. **g** Stability of electroluminescence (EL) of the QLEDs driven at a constant current density of 100 mA cm$^{-2}$.

We demonstrate that electrons but not holes injected into the devices are responsible for the decrease of photoluminescence efficiency of the as-synthesized CdSe/CdS-Cd(RCOO)$_2$ QDs under positive bias (>1.2 V), supporting the picture of operando reduction of cadmium-carboxylate ligands. Single-carrier devices are fabricated to distinguish the effects of the two types of charge carriers. The results (Fig. 2c) indeed show that photoluminescence efficiency of the CdSe/CdS-Cd(RCOO)$_2$ QDs drops in an electron-only device operating at a constant current density of 10 mA cm$^{-2}$ and remains to be constant in a hole-only device operating at a constant current density of 30 mA cm$^{-2}$. As

expected, photoluminescence efficiency of the surface-treated CdSe/CdS-RNH$_2$ QDs is stable in either an electron-only device or a hole-only device (Fig. 2c).

We conduct voltammetric measurements on the surface-treated CdSe/CdS-RNH$_2$ QDs, free cadmium carboxylates, and the as-synthesized CdSe/CdS-Cd(RCOO)$_2$ QDs in anhydrous tetrahydrofuran (Fig. 2d), verifying the potential alignment documented in literature[30,31]. The surface-treated CdSe/CdS-RNH$_2$ QDs exhibit a defined cathodic peak at ~0.89 V (vs normal hydrogen electrode (NHE)), corresponding to their lowest unoccupied molecular orbital (LUMO)[30,31]. In comparison, the

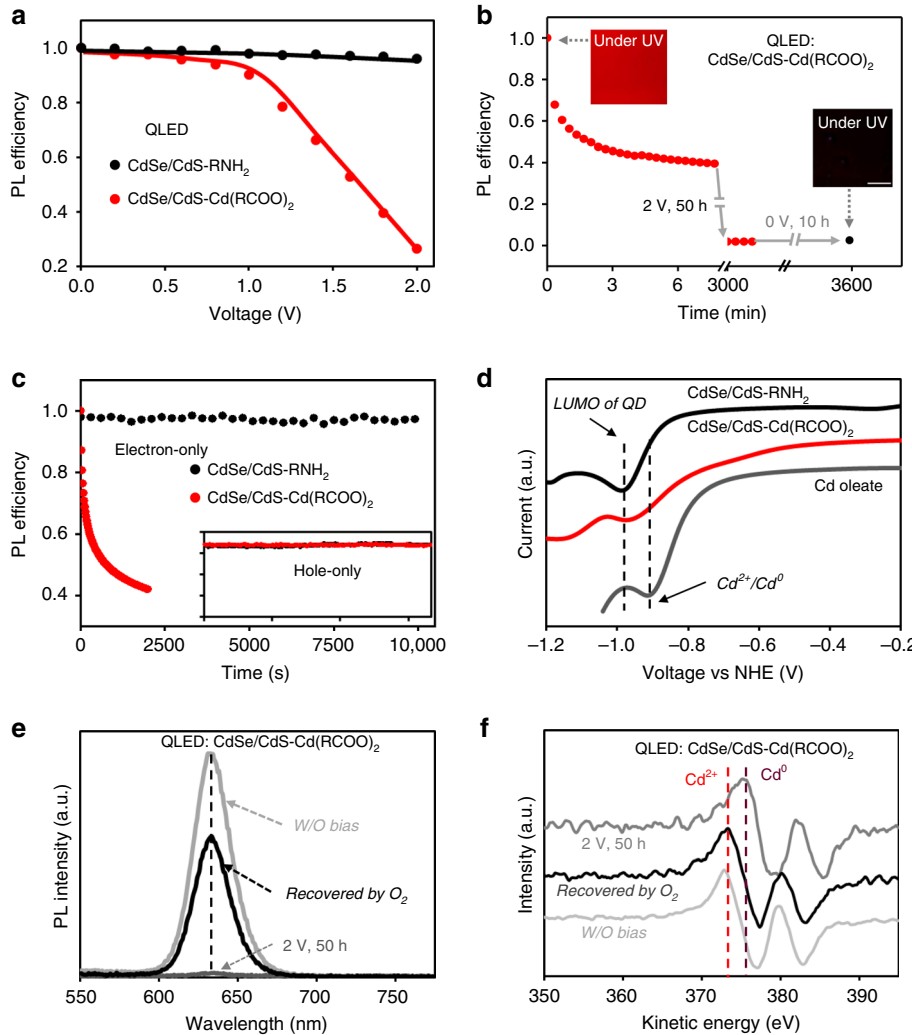

**Fig. 2 Operando electrochemical reduction of the CdSe/CdS-Cd(RCOO)$_2$ QDs. a** Relative PL efficiency of the QDs in the QLEDs as a function of the driving voltage. **b** Relative PL efficiency of the CdSe/CdS-Cd(RCOO)$_2$ QDs in a QLED (driven at a constant voltage of 2 V for 50 h, followed by 0 V for 10 h). Insets are the corresponding photographs of the devices under ultraviolet irradiation (scale bar: 0.5 mm). **c** Relative PL efficiency of the QDs in electron-only devices (ITO/2,2',2"-(1,3,5-Benzinetriyl)-tris(1-phenyl-1-H-benzimidazole) (TPBi, ~25 nm)/QDs (~40 nm)/Zn$_{0.9}$Mg$_{0.1}$O (~60 nm)/Ag, driven at a constant current density of 10 mA cm$^{-2}$). Inset, the relative PL efficiency of the QDs in hole-only devices (ITO/PEDOT:PSS (~35 nm)/poly-TPD (~30 nm)/PVK (~5 nm)/QDs (~40 nm)/4,4'-Bis(N-carbazolyl)-1,1'-biphenyl (CBP, ~25 nm)/MoO$_x$/Au) sharing the same coordinate axes with the main plot. The hole-only devices are driven at a constant current density of 30 mA cm$^{-2}$ to ensure that only holes and no electrons are injected into the devices. **d,** The voltammetric curves of the two types of CdSe/CdS core/shell QDs and cadmium oleate. **e** and **f**, PL and differential auger electron spectra of the CdSe/CdS-Cd(RCOO)$_2$ QDs in QLEDs under different conditions. The top electrodes are delaminated by adhesive tapes and the oxide electron-transporting layers are etched by a dilute acetonitrile solution of acetic acid, a known inert solution for the QD layer.

reduction peak potential for cadmium carboxylates is less negative, agreeing with the literature[23]. The characteristics of the as-synthesized CdSe/CdS-Cd(RCOO)$_2$ QDs exhibit combined features of cadmium carboxylates and the CdSe/CdS-RNH$_2$ QDs. These data indicate that the cadmium-carboxylate ligands should be electrochemically active prior to electron injection into the LUMO of the CdSe/CdS core/shell QDs.

Finally, we confirm that the product of the operando reduction reaction is Cd$^0$. A QLED with as-synthesized CdSe/CdS-Cd (RCOO)$_2$ QDs is biased for ~50 h at 2 V. The top electrode and the electron-transporting layer of this device is removed to expose the QD layer (see captions of Fig. 2e, f for details). While the QDs exposed to nitrogen remain to be non-emissive, their photoluminescence efficiency can be largely recovered after placing them in oxidative environments, either a solution of dibenzoyl peroxide or oxygen gas (Fig. 2e). Importantly, the photoluminescence spectrum of the recovered QDs is identical to that of the

QDs before applying bias stress (Fig. 2e). The QDs subjected to the reduction (in QLED)-oxidation (in oxygen) cycle are further studied by Auger electron spectroscopy, which is sensitive to surface atoms (attenuation length < 1.0 nm)[32] and can readily distinguish the cadmium valence states (Cd$^0$ and Cd$^{2+}$)[33]. Results (Fig. 2f and Supplementary Fig. 5) illustrate that the cadmium species on the QDs after the bias stress at 2 V show signals of Cd$^0$. Conversely, the cadmium species on the other samples of the QDs —either before the bias stress or after the oxidation by oxygen— are Cd$^{2+}$.

**Electroluminescence of QDs by gradual ligand replacement.** Infrared spectra (Fig. 3a) show that the surface treatment outlined in Fig. 1a would completely replace all carboxylate and cadmium-carboxylate ligands by fatty amine ones[22]. As reported in the literature, polar surfaces of zinc-blende II-VI QDs (e.g., {100} and

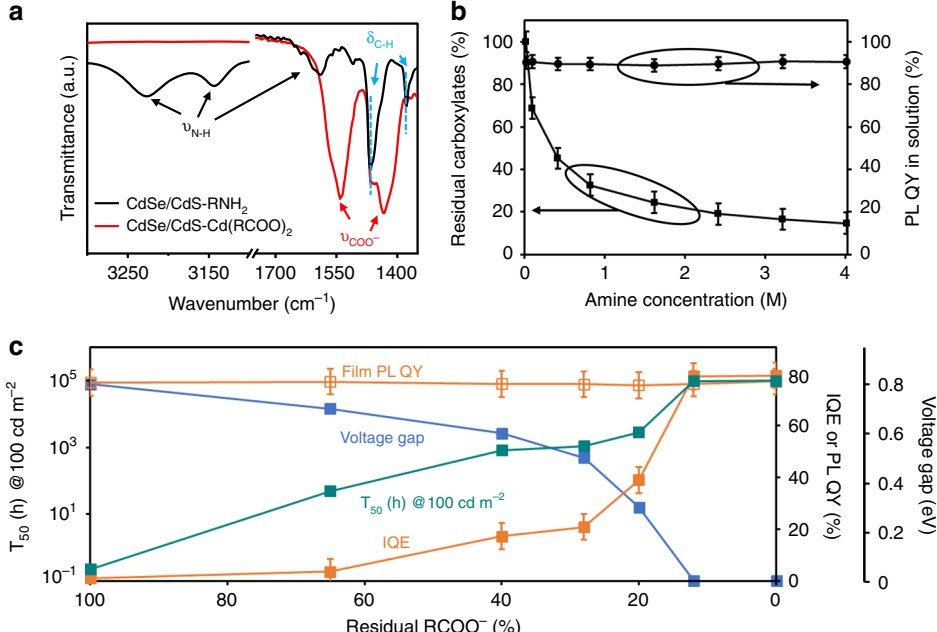

**Fig. 3 Cadmium-carboxylate concentration-dependent PL and EL properties of the CdSe/CdS QDs. a** FTIR spectra of the as-synthesized CdSe/CdS-Cd (RCOO)$_2$ QDs (red) and the completely surface-treated CdSe/CdS-RNH$_2$ QDs (black). Vibration bands of RNH$_2$ and Cd(RCOO)$_2$ are labeled as $\nu_{N-H}$ and $\nu_{COO^-}$, respectively. **b** Residual carboxylate percentages and PL QY of the CdSe/CdS core/shell QDs in solution vs. different concentration of primary amine used in the ligand-exchange procedures. **c** PL QY of the QD films, IQE of the QLEDs, operational lifetime of the QLEDs, and voltage gap of the QLEDs vs. the residual percentage of carboxylate ligands identified by FTIR.

{111} facets) are usually terminated with positively charged cationic sites (i.e., surface cadmium ions for CdSe and CdS lattice), which are coordinated with negatively charged carboxylate ligands to fulfill their four coordination bonds and maintain charge neutrality[20,21,34–36]. The non-polar surfaces ({110} facets as the representative ones) are charge-neutral, on which only neutral cadmium-carboxylate ligands can be weakly adsorbed[20,21,34–36]. The surface Cd cations in the former case are a part of the (surface) lattice, and the ones in the latter case should be more like Cd$^{2+}$ ions in the cadmium-carboxylate molecules[21]. We apply an amine-washing procedure (Fig. 3b), which is milder than the amine-based surface treatment, to gradually replace the carboxylate and cadmium-carboxylate ligands on CdSe/CdS-Cd(RCOO)$_2$ QDs. Bonding between the lattice cadmium sites of the nanocrystals and the carboxylate ligands is about one order of magnitude stronger than that between the neutral nonpolar facets and the adsorbed cadmium-carboxylate ligands[20,21]. Thus, the washing procedure selectively removes the weakly bonded cadmium-carboxylate ligands. The residual carboxylate groups at the plateau in Fig. 3b (~15% of the original ones) should be the tightly bonded carboxylate ones on the polar facets.

Figure 3b reveals that the photoluminescence quantum yield of the QDs with different percentages of the residual carboxylates remains to be near unity in solution, which results in a constant photoluminescence quantum yield of ~80% for all QD thin films (Fig. 3c). However, internal quantum yield and operational lifetime (shown in logarithm scale) of electroluminescence for the series of QLEDs demonstrate a strong dependence on ligand composition of the QDs (Fig. 3c and Supplementary Fig. 6). Simultaneously, the gap between the current-raise voltage and the turn-on voltage is also gradually reduced (Fig. 3c). This voltage gap always co-exists with the operando reduction of Cd$^{2+}$ (or Zn$^{2+}$ below) ions, implying the early current-raise being associated with the operando reduction reactions.

In summary, the results in Fig. 3c confirm that the photoluminescence-electroluminescence gap —both efficiency

and operation lifetime—of the CdSe/CdS core/shell QDs is gradually bridged by removing the weakly adsorbed cadmium-carboxylate ligands. Because the plateau portion of the residual carboxylates (~15%) does not contribute to the photoluminescence-electroluminescence gap, the poor performance of the QLEDs based on the as-synthesized CdSe/CdS-Cd (RCOO)$_2$ QDs should be a result of the operando reduction of the free Cd$^{2+}$ ions in the form of cadmium carboxylates. Below, if similar amine-based surface treatments are applied, we would not specify the small amount of negatively charged carboxylate ligands due to their electrochemically benign nature.

**Ligand-dependent electroluminescence of other red QDs**. The outer shells of core/shell QDs are a part of their inorganic-ligand interface. Thus, the outer shells and their specific ligands might both play a role in determining electrochemical properties of the QDs in QLEDs. Besides CdS, ZnS is the other type of most common outer shells of the colloidal core/shell QDs for electroluminescence applications[14,37,38]. Similar to the cadmium-carboxylate ligands for the CdS outer shells, zinc fatty acid salts (Zn(RCOO)$_2$) are common metal-carboxylate ligands for the ZnS outer shells[28]. Because the standard reduction potential of Zn$^{2+}$ is only ~0.3 V more negative than that of Cd$^{2+}$ (Supplementary Fig. 2), the zinc-carboxylate ligands may also be electrochemically active in QLEDs. In this regard, we adopt another model system for studying the ligand-induced photoluminescence-electroluminescence gap of red-emitting QLEDs, namely, CdSe/CdS/ZnS core/shell/shell QDs. In comparison with the CdSe/CdS core/shell QDs studied above, these core/shell/shell QDs possess the same CdSe core but different shell structure, i.e., five monolayers of CdS inner shells and three monolayers of ZnS outer shells. During the epitaxial growth of the ZnS outer shells, Zn(RCOO)$_2$ and fatty acids are the only source of possible ligands for the resulting QDs (see the "Methods" section).

Results in Fig. 4a–d confirm that a significant photoluminescence-electroluminescence gap exists for the as-synthesized CdSe/CdS/ZnS

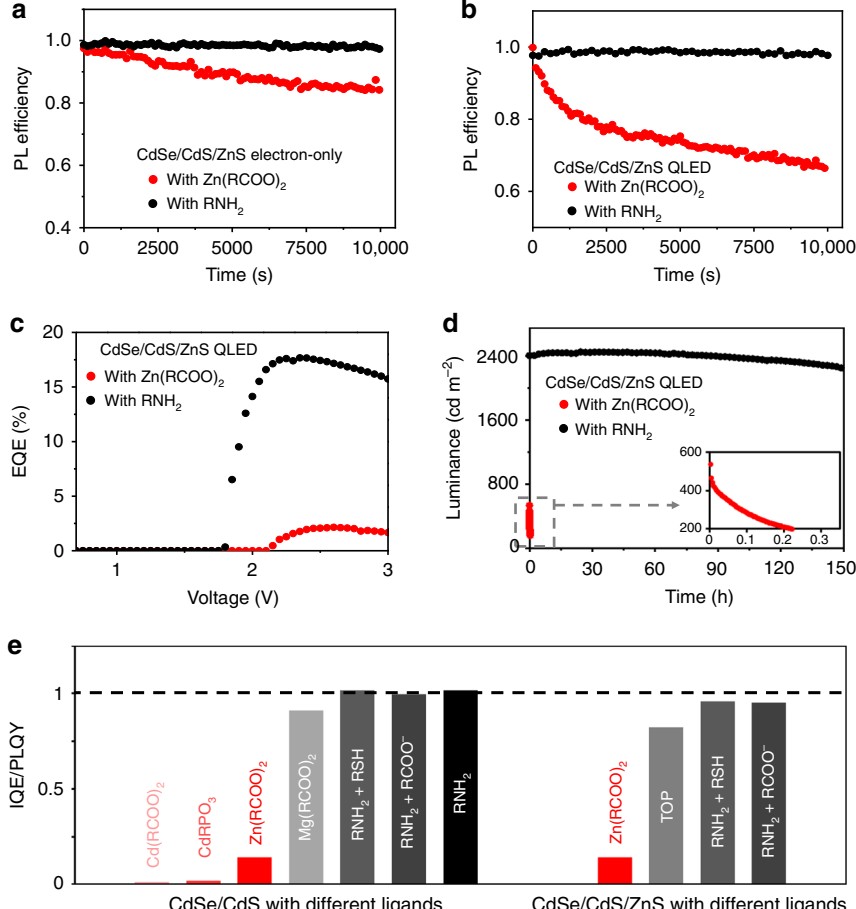

**Fig. 4 Electrochemical stability and QLED performance of the red-emitting QDs with different ligands. a** Relative PL efficiency of the CdSe/CdS/ZnS core/shell/shell QDs in electron-only devices (ITO/TPBi (~25 nm)/QDs (~40 nm)/Zn$_{0.9}$Mg$_{0.1}$O (~60 nm)/Ag). Zn-carboxylate ligands are less reactive than Cd-carboxylate ones, and thus a constant current density of 100 mA cm$^{-2}$ is applied for the electron-only devices, instead of 10 mA cm$^{-2}$ used for the devices with the CdSe/CdS-Cd(RCOO)$_2$ QDs. Inset, the relative PL efficiency of the QDs in hole-only devices (ITO/PEDOT:PSS (~35 nm)/poly-TPD (~30 nm)/PVK (~5 nm)/QDs (~40 nm)/CBP (~25 nm)/MoO$_x$/Au) driven at a constant current density of 30 mA cm$^{-2}$, sharing the same coordinate axes with the main plot. **b**, Relative PL efficiency of the CdSe/CdS/ZnS core/shell/shell QDs in QLEDs driven at a constant current density of 100 mA cm$^{-2}$. **c** EQE vs. driving voltage characteristics and **d**, Stability data of the QLEDs based on the CdSe/CdS/ZnS core/shell/shell QDs driven at the constant-current mode. **e** The IQE/PL QY ratio (IQE of the QLED divided by PL QY of the QD film) for the CdSe/CdS core/shell and the CdSe/CdS/ZnS core/shell/shell QDs with different ligands.

core/shell/shell QDs with zinc-carboxylate ligands (CdSe/CdS/ZnS-Zn(RCOO)$_2$). This gap is removed for the surface-treated CdSe/CdS/ZnS core/shell/shell QDs with amine ligands (CdSe/CdS/ZnS-RNH$_2$ QDs). Both the CdSe/CdS/ZnS-Zn(RCOO)$_2$ and the CdSe/CdS/ZnS-RNH$_2$ QDs possess the same photoluminescence quantum yield of ~80% in thin films. The relative photoluminescence efficiency of the as-synthesized CdSe/CdS/ZnS-Zn(RCOO)$_2$ QDs in both working QLEDs and electron-only devices decreases gradually (Fig. 4a, b), while the surface-treated CdSe/CdS/ZnS-RNH$_2$ QDs offer stable and efficient photoluminescence in the working devices. The QLEDs using the as-synthesized CdSe/CdS/ZnS-Zn(RCOO)$_2$ QDs show low average EQEs of 2.7 ± 0.5% and negligible device lifetime, but the surface-treated CdSe/CdS/ZnS-RNH$_2$ QDs demonstrate high-performance QLEDs with an average EQE of 17.8 ± 0.4% (Fig. 4c) and excellent operational lifetime, i.e., a typical T$_{95}$ lifetime (time for the luminance decreasing to 95% of the original value) of ~600 h at 1000 cd m$^{-2}$ (Fig. 4d).

In addition to the ligand systems discussed above, there are some other common types of ligands for colloidal QDs, including organophosphines[5,10,15], metal phosphonates[39,40], and thiols[2]. Figure 4e illustrates the photoluminescence-electroluminescence efficiency comparison for both CdSe/CdS core/shell and CdSe/

CdS/ZnS core/shell/shell QDs with different ligands. Table 1 includes the absolute photoluminescence quantum yield of the QDs films, electroluminescence efficiency, and lifetime of the QLEDs. The neutral cadmium-phosphonate ligands cause a very large photoluminescence-electroluminescence gap, similar to that induced by the cadmium-carboxylate ligands (raw data shown in Supplementary Fig. 7a, b). Thiol ligands significantly quench both photoluminescence and electroluminescence of the CdSe/CdS-Cd(RCOO)$_2$ QDs (Supplementary Fig. 7a, b) but barely affect the photoluminescence and electroluminescence of CdSe/CdS-RNH$_2$ QDs (Supplementary Fig. 7c–f), implying detrimental effects of the neutral cadmium-thiolate ligands but electrochemically benign nature of the thiolate ligands[41]. The effects of neutral tri-octylphosphine (TOP) ligands are quite similar to those of the amine ones in terms of eliminating the photoluminescence-electroluminescence gap (raw data shown in Supplementary Fig. 7c–f). Replacing cadmium carboxylates by zinc carboxylates as ligands results in a mild reduction of the photoluminescence-electroluminescence gap for the CdSe/CdS QDs. Magnesium-carboxylate (Mg(RCOO)$_2$) ligands largely eliminate the photoluminescence-electroluminescence gap of the resulting QLEDs (Supplementary Fig. 7g, h), which are

**Table 1 PL and EL performance of the QDs with various combinations of shells and ligands.**

| Inorganic structure | Ligands | Film PL QY (%) | EQE (%) | Device lifetime[a] $T_{50}$ (h) |
|---|---|---|---|---|
| CdSe/CdS | Cd(RCOO)$_2$ | 77 ± 2 | 0.25 ± 0.08 | ~0.3 |
| | Cd(RPOOO) | 70 ± 3 | 0.1–0.3 | ~1 |
| | Zn(RCOO)$_2$ | 75 ± 2 | 2–4 | ~5–7 |
| | Mg(RCOO)$_2$ | 42 ± 3 | 8–9 | ~8000 |
| | RNH$_2$ + RSH | 77 ± 2 | 18–20 | ~90,000 |
| | RNH$_2$[b] | 76 ± 2 | 18.5 ± 0.6 | ~90,000 |
| | RNH$_2$ | 76 ± 2 | 18.6 ± 0.6 | ~90,000 |
| CdSe/CdS/ZnS | Zn(RCOO)$_2$ | 80 ± 2 | 2.7 ± 0.5 | ~3–5 |
| | TOP[c] | 79 ± 2 | 15.4 ± 0.6 | $T_{95}$ @1000 nits ~400 |
| | RNH$_2$[b] | 79 ± 2 | 17.8 ± 0.4 | $T_{95}$ @1000 nits ~600 |
| | RNH$_2$ + RSH[b] | 78 ± 2 | 17–19 | $T_{95}$ @1000 nits ~550 |
| CdSe/CdZnSe/CdZnS | Zn(RCOO)$_2$ | 74 ± 2 | 5–7 | $T_{95}$ @1000 nits ~80 |
| | TOP[c] | 75 ± 2 | 15.8 ± 0.9 | $T_{95}$ @1000 nits ~3800 |

The data shown in italics are from QDs with electrochemically reactive ligands and the rest of the data are from QDs with electrochemically inert ligands.
[a]The device lifetimes are estimated at an initial brightness of 100 nits if not noted.
[b]These QDs are determined to retain their negatively charged carboxylate ligands (~15% of residual carboxylate ligands).
[c]The percentage of carboxylate ligands of these QDs after ligand exchange by TOP was ~25%. This value is the lower limit for maintaining the dispersity of these QDs after TOP treatment.

consistent with the high reduction potential of Mg$^{2+}$ ions (Supplementary Fig. 2).

In addition to CdS and ZnS, zinc and cadmium chalcogenide alloys are sometimes applied as outer shells. Experimental results based on the CdSe/CdZnSe/CdZnS core/shell/shell QDs illustrate that a significant photoluminescence-electroluminescence gap exists when zinc-carboxylate ligands are in place (Table 1 and Supplementary Fig. 8). Replacement of the neutral zinc-carboxylate ligands by TOP ligands leads to devices exhibiting an extremely long operation $T_{95}$ lifetime of 3800 h at 1000 cd m$^{-2}$ (Table 1 and Supplementary Fig. 8), surpassing the stability record of QLEDs[15].

In summary, above results (Fig. 4e, Table 1, Supplementary Fig. 7, and Supplementary Fig. 8) suggest that similar to fatty amines, neutral organophosphines are electrochemically inert ligands for the core/shell QDs used in QLEDs. For the ligands of Cd/Zn-based salts, no matter what anionic groups (phosphonates, carboxylates, or thiolates) are, they are all detrimental to performance of the QLEDs. From cadmium carboxylates, zinc carboxylates, to magnesium carboxylates, the detrimental effects to electroluminescence of the CdSe/CdS QDs alleviate gradually. All common types of semiconductor outer shells are electrochemically inert in the QLEDs as long as electrochemically inert ligand systems are in place. All these results, including both electroluminescence efficiency and lifetime, are consistent with the above conclusion, that the degradation channel induced by the operando electrochemical reactions of surface ligands is critical for the performance of QLEDs.

**Improving the operational lifetime of blue-emitting QLEDs**. The short operational lifetime of blue-emitting QLEDs is currently the main bottleneck for commercialization of the QLED technology. Blue-emitting CdSeS/ZnSeS/ZnS core/shell/shell QDs with either zinc-carboxylate (CdSeS/ZnSeS/ZnS-Zn(RCOO)$_2$ QDs) or amine ligands (CdSeS/ZnSeS/ZnS-RNH$_2$ QDs) are comparatively studied.

Photoluminescence of the CdSeS/ZnSeS/ZnS-Zn(RCOO)$_2$ QDs under electric bias is unstable in working electron-only devices and QLEDs (Fig. 5a, b), indicating similar electrochemical degradation discussed above. Conversely, photoluminescence of the CdSeS/ZnSeS/ZnS-RNH$_2$ QDs under electric bias is stable during the operation of both electron-only devices and QLEDs (Fig. 5a, b). Both electroluminescence efficiency and device operation lifetime of the QLEDs with the CdSeS/ZnSeS/ZnS-Zn(RCOO)$_2$ QDs are poor (Fig. 5c, d). In comparison, blue-emitting

QLEDs with the CdSeS/ZnSeS/ZnS-RNH$_2$ QDs show decent efficiency, as characterized by a peak EQE of ~10% (Fig. 5c), though their photoluminescence quantum yield in thin film (~60%) is lower than that (~80%) of the CdSeS/ZnSeS/ZnS-Zn(RCOO)$_2$ QDs (Supplementary Fig. 9). The operational lifetime of the QELDs with the CdSeS/ZnSeS/ZnS-Zn(RCOO)$_2$ QDs is also substantially enhanced (~85 h at an initial brightness of 1450 cd m$^{-2}$) (Fig. 5d). By applying an experimentally determined acceleration factor of 1.88 (Supplementary Fig. 9), our devices possess a lifetime of >10,000 h at 100 cd m$^{-2}$, representing the most stable blue QLED reported so far.

**General trends and new strategy**. In literature, QLEDs with high efficiencies—sometimes also with decent device lifetime—have been reported by several groups using the core/shell QDs[9,10,15–17,42–47]. In all these reports (Supplementary Table 1), a common feature is that excessive electrochemically inert ligands identified above, i.e., fatty amines and/or organophosphines, are used in the synthetic systems. It is known that, in such a system, the neutral cadmium/zinc-carboxylate ligands can be largely removed during isolation of the QDs[20,34,48], which would result in unintentional fulfillment of the criterion for electrochemically stable QDs for QLEDs. We emphasize that, without full control of surface ligands of QDs by design, fabricating QLEDs with high efficiency, long lifetime, and good reproducibility would remain to be empirical.

It is interesting to note that electrochemically inert ligands can only bridge the photoluminescence-electroluminescence gap—both efficiency and lifetime—of the QDs, and high efficiency of photoluminescence and electroluminescence requires a suited combination of QD structures and electrochemically inert ligands. For instance, with the ZnSe inner shells and thin ZnS outer shells—generally true for the blue-emitting QLEDs in Fig. 5 and those non-cadmium InP QLEDs[49], removal of zinc carboxylate ligands by either amines or phosphines is less controllable and sometimes reduces photoluminescence efficiency (Supplementary Fig. 9), which would counteract the improvement of device performance by the electrochemically inert ligands. Thus, design of new core/shell structures coupled with further development of electrochemically inert ligands, including the surface-treatment procedures, is the direction to pursuit for achieving ideal performance of various QLEDs. In this regard, though they are under-investigated, magnesium fatty acid salts (also other electrochemically stable metal ions) are interesting candidates for broadening the spectrum of electrochemically inert ligands (Fig. 4e and Table 1).

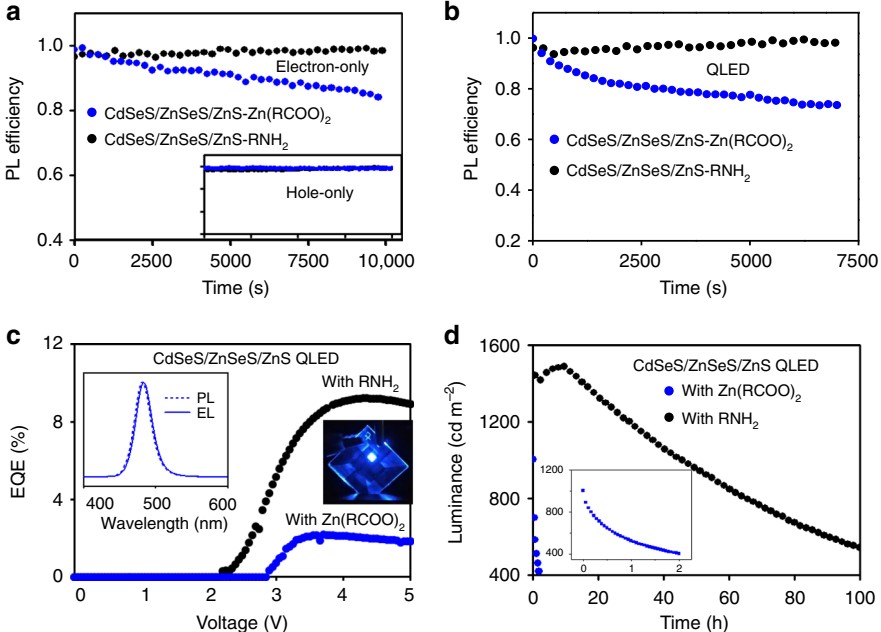

**Fig. 5 Electrochemical stability and QLED performance of the blue-emitting QDs. a** Relative PL efficiency of the QDs in the electron-only devices (ITO/TPBi (~25 nm)/QDs (~20 nm)/$Zn_{0.9}Mg_{0.1}O$ (~50 nm)/Al) driven at a constant current density of 100 mA $cm^{-2}$. Inset, the relative PL efficiency of the QDs in the hole-only devices (ITO/PEDOT:PSS (~35 nm)/poly-TPD (~30 nm)/PVK (~5 nm)/QDs (~20 nm)/CBP (~25 nm)/$MoO_x$/Au) driven at a constant current density of 30 mA $cm^{-2}$, which shares the same coordinate axes with the main plot. **b** Relative PL efficiency of the QDs in the QLEDs (ITO/PEDOT:PSS (~35 nm)/poly-TPD (~30 nm)/PVK (~5 nm)/QDs (~20 nm)/$Zn_{0.9}Mg_{0.1}O$ nanocrystals (~50 nm)/Al, driven at a constant current density of 100 mA $cm^{-2}$). **c** EQE vs. driving voltage of the blue-emitting QLEDs. Insets, PL and EL spectra and a photograph of a working QLED, respectively. **d** Stability data of the blue-emitting QLEDs driven at the constant-current mode. Inset: expanded plot for the QLEDs with the CdSeS/ZnSeS/ZnS-Zn(RCOO)2 QDs.

## Discussion

In conclusion, it is indeed interesting to see that a simple, general, and materials-based strategy can effectively bridge the photoluminescence-electroluminescence gap of QDs, leading to red-emitting and blue-emitting QLEDs with record-long operational lifetime. These advancements lead us to believe that there are no fundamental obstacles to fully exploit the readily achievable superior photoluminescence properties of QDs to realize high-performance and solution-processed electroluminescence devices, such as QLEDs, quantum light sources, and lasers. Our study further suggests that the electrochemical stability of ligands is a critical design parameter for QDs used in optoelectronic and electronic devices because operation of these devices all involves charge injection into the QDs.

## Methods

**Materials**. Poly-TPD (average molecular weight, ~60,000 g $mol^{-1}$) was purchased from Xi'an polymer light technology Corp. PVK (average molecular weight, 25,000–50,000 g $mol^{-1}$), sulfur (99.98%), zinc acetate dihydrate (98%), 2-ethylhexanethiol (97%) and oleylamine (70%) were purchased from Sigma Aldrich. Octylamine (98%), tri-octylphosphine (TOP, 97%), cadmium acetate (98%), magnesium acetate hydrate (98%), benzoyl peroxide (BPO, 97%, dry wt.), 1-octadecene (ODE, 90%) and oleic acid (90%) were purchased from Alfa-Aesar. Chlorobenzene (extra dry, 99.8%), m-xylene (extra dry, 99%), octane (extra dry, >99%), ethanol (extra dry, 99.5%), tetrahydrofuran (THF, 99.9%, anhydrous) and decanoic acid (99%) were purchased from Acros. Dimethyl sulphoxide (DMSO, HPLC grade) and ethyl acetate (HPLC grade) were purchased from J&K Chemical Ltd. 1-Ethyl-3-methylimidazolium Bis(trifluoromethanesulfonyl)imide (EMI-TFSI, 98%) and Ferrocene (98%) were purchased from Energy Chemical. Red CdSe/CdZnSe/CdZnS core/shell/shell QDs and blue CdSeS/ZnSeS/ZnS core/shell/shell QDs were purchased from Najing technology Co., Ltd.

**Syntheses of QDs and oxide nanocrystals**. The CdSe/CdS core/shell QDs (ten monolayers of CdS shell) were synthesized according to ref. [14] with some modifications. Briefly, core CdSe dots (first exciton peak: 550 nm) were used for shell growth. ODE (4 mL), oleic acid (0.85 mL), decanoic acid (0.15 mL), and cadmium acetate (1 mmol) were mixed and degassed at 150 °C for 10 min, followed by

injecting of a hexane solution (1 mL) containing the CdSe core QDs. The mixture was further degassed for 10 min. Then the temperature of the solution was raised to 260 °C. A solution of sulfur (0.5 mmol) in ODE (5 mL) was dropwisely introduced into the reaction flask at a rate of 4 mL/h. After 60 min, the reaction was stopped. The QDs were precipitated by adding acetone and then purified twice by using toluene and methanol for dissolution and precipitation, respectively. The resulting CdSe/CdS core/shell QDs, i.e., CdSe/CdS-Cd(RCOO)2 QDs, were dissolved in toluene.

For the synthesis of CdSe/CdS/ZnS core/shell/shell QDs (five monolayers of CdS shell and three monolayers of ZnS shell), the CdSe/CdS core/shell QDs (five monolayers of CdS shell) were synthesized and precipitated following the above procedures. For the growth of ZnS shell, ODE (4 mL), oleic acid (0.4 mL), decanoic acid (0.1 mL), and zinc acetate (0.5 mmol) were mixed and degassed at 150 °C for 10 min, followed by injecting of a hexane solution (1 mL) containing CdSe/CdS QDs. The mixture was further degassed for 10 min. Then the temperature of the solution was raised to 300 °C. A solution of sulfur (0.5 mmol) in ODE (5 mL) was dropwisely introduced into the reaction flask at a rate of 4 mL/h. After 30 min, the reaction was stopped. The QDs were precipitated by adding acetone and then purified twice by using toluene and methanol for dissolution and precipitation, respectively. The resulting CdSe/CdS/ZnS core/shell/shell QDs, i.e., CdSe/CdS/ZnS-Zn(RCOO)2 QDs were dissolved in toluene.

The $Zn_{0.9}Mg_{0.1}O$ nanocrystals were synthesized according to ref. [9]. For a typical synthesis, a DMSO solution (30 mL) of magnesium acetate hydrate (0.3 mmol) and zinc acetate hydrate (2.7 mmol) mixed dropwise with an ethanol solution (10 mL) of TMAH (5 mmol) and stirred for 1 h in ambient conditions. The resulting $Zn_{0.9}Mg_{0.1}O$ nanocrystals were precipitated by adding ethyl acetate and dispersed in ethanol. The ethanol solution of $Zn_{0.9}Mg_{0.1}O$ nanocrystals (~30 mg $mL^{-1}$) was filtered with 0.22 μm PTFE filter before use.

**Surface modification of the QDs**. The as-synthesized CdSe/CdS-Cd(RCOO)2 QDs were dissolved in a mixture of oleylamine (2 mL), sulfur (0.1 mmol) and ODE (1 mL). The temperature was raised to 110 °C and maintained for 20 min, followed by precipitation of QDs by adding methanol (5 mL). The procedures were repeated to ensure complete conversion of the surface ligands. The precipitated QDs were dissolved in ODE (2 mL) and oleylamine (2 mL) and then annealed at 240 °C for 20 min. The resulting CdSe/CdS-RNH2 QDs were precipitated by adding methanol.

Regarding the ligand exchange of CdSe/CdS-Cd(RCOO)2 QDs by amine, the as-synthesized CdSe/CdS-Cd(RCOO)2 QDs were dispersed in a mixture of oleylamine, octylamine and ODE, with a total volume of 6 mL. The volume ratio of oleylamine and octylamine was maintained at 3: 2 and the total concentrations of the amines were varied from 0-4 mol/L. The mixture was heated up to 150 °C and

annealed for 20 min. Then the solution was cooled. The QDs were precipitated by adding methanol (12 mL). Next, the QDs were reprecipitated by using 1 mL hexane and 8 mL acetonitrile.

Regarding the ligand exchange of CdSe/CdS/ZnS-Zn(RCOO)$_2$ QDs by amine, the as-synthesized QDs were dispersed in a mixture of oleylamine (3.6 mL) and octylamine (2.4 mL). The mixture was heated to 50 °C and annealed for 20 min. Then the solution was cooled. The QDs were first precipitated by adding methanol (12 mL) and then reprecipitated by using 1 mL hexane and 8 mL acetonitrile. The number of the ligand-exchange cycle was repeated four times, leading to CdSe/CdS/ZnS-RNH$_2$ QDs.

Regarding the ligand exchange of CdSeS/ZnSeS/ZnS-Zn(RCOO)$_2$ QDs by amine, the as-synthesized QDs were dispersed in a mixture of oleylamine (1 mL) and ODE (2 mL). The mixture was heated to 220 °C and annealed for 10 min. Then the solution was cooled, leading to CdSeS/ZnSeS/ZnS-RNH$_2$ QDs. The QDs were precipitated by adding methanol (6 mL).

Regarding the CdSe/CdS/ZnS-TOP QDs, the ligand-exchange procedure was similar to that of amine for CdSe/CdS/ZnS-Zn(RCOO)$_2$ QDs, except that pure TOP (6 mL) was used instead of oleylamine and octylamine.

Regarding the CdSe/CdZnSe/CdZnS-TOP QDs, the ligand-exchange procedure was similar to that for CdSe/CdS/ZnS-TOP QDs, except that the CdSe/CdZnSe/CdZnS-Zn(RCOO)$_2$ QDs were used.

The ligand-exchange procedure with 2-ethylhexanethiol was performed by adding 1 mL thiol into 1 mL solution of QDs (either the CdSe/CdS-RNH$_2$ QDs or the CdSe/CdS/ZnS-RNH$_2$ QDs) and then stirred for 1 h in N$_2$ atmosphere at 70 °C. After cooling of the mixture, the QDs were precipitated by adding ethanol and re-dispersed in octane.

Regarding the CdSe/CdS-Cd(RPOOO) QDs, the CdSe/CdS-Cd(RCOO)$_2$ QDs were dispersed in toluene (2 mL). Octylphosphonic acid (0.25 mmol) was added into the mixture. The solution was heated to 50 °C and maintained for 1 h. Then ethanol was added to precipitate the QDs.

Regarding the CdSe/CdS-Zn(RCOO)$_2$ QDs, ODE (4 mL), oleic acid (0.4 mL), decanoic acid (0.1 mL), and zinc acetate (0.5 mmol) were mixed and degassed at 150 °C for 10 min, followed by injecting of a hexane solution (1 mL) containing CdSe/CdS-RNH$_2$ QDs. The mixture was further degassed for 10 min. Then the temperature of the solution was raised to 200 °C. After annealed for 10 min, the reaction was stopped. The QDs were precipitated by adding acetone and then purified twice by using toluene and methanol for dissolution and precipitation, respectively. The resulting CdSe/CdS-Zn(RCOO)$_2$ QDs were dissolved in toluene.

Regarding the CdSe/CdS-Mg(RCOO)$_2$ QDs, the procedure was similar to that for CdSe/CdS-Zn(RCOO)$_2$ QDs except that zinc acetate was replaced by magnesium acetate.

The relative amount of the residue carboxylate ligands of QDs were determined by Fourier transform infrared spectra. The samples were prepared by dispersing the QDs in dodecane, followed by adding hydrochloric acid. The mixture was vortexed for 10 min to convert the carboxylate ligands (including both metal carboxylate sales and carboxylate ions) to carboxylic acid. Next, the mixture was centrifuged for 2 min at 4000 rpm. The concentration of carboxylic acid in the supernatant was determined by a standard curve (Supplementary Fig. 6a).

**Device fabrication.** The QLEDs were fabricated by depositing materials onto ITO coated glass substrates (~1.1 mm in thickness, sheet resistance: ~20 Ω sq$^{-1}$). PEDOT:PSS solutions (Baytron P VP Al 4083, filtered through a 0.22 μm N66 filter) were spin-coated at 3000 rpm for 60 s and then baked at 150 °C for 15 min. The PEDOT:PSS-coated substrates were subjected to an oxygen plasma treatment for 5 min and then transferred into a nitrogen-filed glove box (O$_2$ < 1 ppm, H$_2$O < 1 ppm). Poly-TPD (in chlorobenzene, 8 mg mL$^{-1}$), PVK (in m-xylene, 1.5 mg mL$^{-1}$), QDs (in octane, ~20 mg mL$^{-1}$ for the CdSe/CdS core/shell red dots, ~15 mg mL$^{-1}$ for the CdSe/CdS/ZnS core/shell/shell red dots and the CdSe/CdZnSe/CdZnS core/shell/shell red dots, ~ 12 mg mL$^{-1}$ for the CdSeS/ZnSeS/ZnS core/shell/shell blue dots), and Zn$_{0.9}$Mg$_{0.1}$O nanocrystals (in ethanol, 30 mg mL$^{-1}$) were layer-by-layer deposited by spin coating at 2000 rpm for 45 s. The poly-TPD and PVK layers were baked at 130 °C for 20 min and at 150 °C for 30 min, respectively, before deposition of the next layer. Finally, metal electrodes (100 nm) were deposited using a thermal evaporator (Trovato 300 C, base pressure: ~2 × 10$^{-7}$ torr) through a shadow mask. The device area was 4 mm$^2$ as defined by the overlapping area of the ITO and metal electrodes. All devices were encapsulated in a glove-box using commercially available ultraviolet-curable resin.

**Characterizations.** The absorption spectra were obtained by using an Agilent Cary 5000 spectrophotometer. The photoluminescence spectra were obtained by using an Edinburgh Instruments FLS920 fluorescence spectrometer. The time-resolved fluorescence spectra were measured by the time-correlated single-photon counting method using an Edinburgh Instruments FLS920 spectrometer. The samples were excited by a 405 nm pulsed diode laser (EPL-405). The absolute photoluminescence quantum efficiencies of the QDs were measured by applying a two-step (for solutions) method or a three-step (for thin films) method[50]. A system consisting of a Xenon lamp, optical fiber, a QE65000 spectrometer (Ocean Optics) and a home-designed integrating sphere was used. Transmission electron microscope observations were carried out using Hitachi 7700 operated at 80 keV. Fourier transform infrared spectroscopy analyses were conducted using a Nicolet 380 spectrometer.

Energy-dispersive X-ray spectroscopy analyses of the QDs were carried out on an Ultral 55 scan transmission electron microscope. Auger electron spectroscopy measurements were performed using a PHI 710 Scanning Auger Microscopy. The thicknesses of the films were measured using a KLA Tencor P-7 Stylus Profiler.

Single-dot photoluminescence experiments were conducted on a home-built epi-illumination fluorescence microscope system equipped with a Zeiss 63× oil immersion objective (N.A.:1.46). Samples were excited by a 405 nm ps pulse laser with 1 MHz repetition for photoluminescence measurement. The photoluminescence intensity trajectories of single dots were recorded by an electron multiplier charge-coupled device (EMCCD, Andor iXon3). The exposure time per frame was 30 ms.

Voltammetric measurements were carried out by using a CHI600D electrochemical workstation located in a glove box. Typically, a solution of THF (1 mL) containing EMI-TFSI (0.1 mmol) was used as an electrolyte and a solution of Ferrocene in THF (0.1 M) was added as a reference. Silver wire and platinum coil were used as quasi-reference and counter electrodes, respectively. Solutions (0.1 mL) of QDs in toluene was added for measurements. The scan speed was 20 mV s$^{-1}$.

All QLEDs were characterized under ambient conditions (room temperature: 22~24 °C and relative humidity: 40~60%). A system consisting of a Keithley 2400 source meter and an integration sphere (FOIS-1) coupled with a QE-Pro spectrometer (Ocean Optics) was used to measure the current density-luminance-voltage curves. The electroluminescence characteristics of the QLEDs were cross-checked at Cavendish Laboratory (Richard Friend group) and Nanjing Tech University (Jianpu Wang group). The operational lifetimes of the QLEDs were measured by using an aging system designed by Guangzhou New Vision Opto-Electronic Technology Co., Ltd.

Photoluminescence intensity changes of the QDs in working devices were measured via a home-build system controlled by Labview. The system (Supplementary Fig. 4) consists of lock-in amplifiers (SR830), source-meter (Keithley 2400), electric-meter (Keithley 2000), and photodetectors (Thorlabs PDA100A). The devices were excited by a cw 445 nm (or 405 nm) laser with an optical chopper for frequency modulation (933 Hz). A Keithley 2400 source-meter was used to electrically drive the devices. Lock-in amplifiers combined with photodetectors were used to independently measure the intensity of photoluminescence from QDs and excitation light. The intensity of excitation light was kept to be low to ensure that the photo-excitation do not interfere with the operation of the devices.

## Data availability
The data that support the finding of this study are available from the corresponding authors upon reasonable request.

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

## Acknowledgements

This work was financially supported by the National Key R&D Program of China (2016YFB0401600), the National Natural Science Foundation of China (91833303), and the Fundamental Research Funds for the Central Universities (2017XZZX001-03A). We thank Prof. Peng Wang and Mr. Yanlei Hao (Zhejiang University, China) for the assistances on providing the access to voltammetric equipment and the fabrication of QLEDs, respectively.

## Author contributions

X.P. and Y.J. conceived the idea, supervised the work, and wrote the manuscript. C.P. participated in conceiving the idea, developed the surface treatment processes for all QDs, and conducted voltammetric measurements. X.D. carried out the fabrication and characterizations of QLEDs and single-carrier devices. Y.S. developed the CdSe/CdS/ZnS core/shell/shell QDs and worked on surface modification of all QDs. M.Z. carried out the single-dot photoluminescence experiments. Y.D. calculated the out-coupling efficiency of the QLEDs. All authors discussed the results and commented on the manuscript.

## Competing interests

The authors declare no competing interests.
