## [Peer Review File · Nature Communications]

Reviewers' comments:

Reviewer #1 (Remarks to the Author):

This manuscript reports a study of the performance of LEDs made with core/shell quantum dots, as a function of stabilizing ligand. This is an extensive, carefully done work of high quality, with much supporting data. The findings seem fully supported to me. The main finding is that in LEDs, dots with amine ligands work very well, and that dots with Cd carboxylate ligands work poorly, although both kinds of dots work equally well in photoluminescence. The authors claim that this establishes a new degradation channel in quantum dot LEDs, namely the possible existence of irreversible electrochemical reaction of a ligand, that has not previously been understood by the community. In these specific LEDs, electrons reduce Cd ions in the carboxylate to Cd atoms which then quench electroluminescence.

Apparently it is true this was not previously recognized. Previous LED studies, ref 15-17 and 42-47, that reported high efficiency just happened to all use ligands that do not undergo ligand electrochemistry under operating conditions, as the authors of this manuscript state. The electrochemical properties of the ligand were not part of the experimental design.

Quantum dot LEDs are of widespread interest presently in the TV display industry. This article will attract broad interest for the chemical insight into the problem.

On page 6 the first sentence, at the top of the page, does not make sense. In some other places, the wording is somewhat awkward. Yet overall the ideas are clearly explained.

Reviewer #2 (Remarks to the Author):

The author demonstrated record-long operational lifetimes for both red-emitting QLEDs ($T_{95} > 3800$ hours at 1000 cd m^{-2}) and blue-emitting QLEDs ($T_{50} > 10,000$ hours at 100 cd m^{-2}). Through a general but previously overlooked degradation channel in QLEDs, i.e., operando electrochemical reactions of surface ligands with injected charge carriers, they developed a strategy of applying electrochemically-inert ligands to QDs with excellent luminescent properties to bridge their photoluminescence electroluminescence gap. This is an interesting working and it is meaningful for the QLED application especially for the commercial application. However; some questions should be addressed before accepted.

1 In Page 7, why take the 10 mA cm^{-2} for testing the CdSe/CdS, and for the CdSe/CdS/ZnS and CdSeS/ZnSeS/ZnS with 100 mA cm^{-2} ? Why use the 30 mA cm^{-2} but not the 100 mA cm^{-2} for the testing?

2 In Page 8, the author demonstrated that completely replace all carboxylate and cadmium-carboxylate ligand, however the $-\text{COO}^-$ peak appeared in the Fig. 3a. How to explain this?

Reviewer #3 (Remarks to the Author):

The paper "Electrochemically-stable ligands bridge the photoluminescence..." is very interesting and well written paper. It emphasizes that the issue that material design requirements for EL applications does not always coincide with what is good enough for PL applications. The device performance is impressive and as far as the device performance and its relation to the presence of different ligands, I find the paper to be rigorous and providing several different indications that converge to the same conclusion. If you choose the ligand wrong – your film PL will have very little in common with its PL.

There is also significant part of the paper that mentions electrochemical activity. Unfortunately, I am not an expert in this. I can't tell if the improved results are due to better coverage and/or

passivation of the surface or that there is indeed an electrochemistry that involves one ligand and not the other. I hope that this important issue can be answered by one of the other referees. Assuming this part is correct than I believe the paper is suitable for publication.

Responses to Reviewer #1

This manuscript reports a study of the performance of LEDs made with core/shell quantum dots, as a function of stabilizing ligand. This is an extensive, carefully done work of high quality, with much supporting data. The findings seem fully supported to me. The main finding is that in LEDs, dots with amine ligands work very well, and that dots with Cd carboxylate ligands work poorly, although both kinds of dots work equally well in photoluminescence. The authors claim that this establishes a new degradation channel in quantum dot LEDs, namely the possible existence of irreversible electrochemical reaction of a ligand, that has not previously been understood by the community. In these specific LEDs, electrons reduce Cd ions in the carboxylate to Cd atoms which then quench electroluminescence.

Apparently it is true this was not previously recognized. Previous LED studies, ref 15-17 and 42-47, that reported high efficiency just happened to all use ligands that do not undergo ligand electrochemistry under operating conditions, as the authors of this manuscript state. The electrochemical properties of the ligand were not part of the experimental design.

Quantum dot LEDs are of widespread interest presently in the TV display industry. This article will attract broad interest for the chemical insight into the problem.

On page 6 the first sentence, at the top of the page, does not make sense. In some other places, the wording is somewhat awkward. Yet overall the ideas are clearly explained.

Our revision and responses: We thank the reviewer for his/her positive comments on the scientific quality of our manuscript. Following the reviewer's suggestions, the first sentence on page 6 (highlighted) has been modified to "In literature, chemical reduction of cadmium-salt ligands on the cadmium chalcogenide QDs was reported^{26,27}". In addition, we have gone through the entire manuscript to correct the spelling, grammar and word use.

Responses to Reviewer #2

The author demonstrated record-long operational lifetimes for both red-emitting QLEDs ($T_{95} > 3800$ hours at 1000 cd m^{-2}) and blue-emitting QLEDs ($T_{50} > 10,000$ hours at 100 cd m^{-2}). Through a general but previously overlooked degradation channel in QLEDs, i.e., operando electrochemical reactions of surface ligands with injected charge carriers, they developed a strategy of applying electrochemically-inert ligands to QDs with excellent luminescent properties to bridge their photoluminescence electroluminescence gap. This is an interesting working and it is meaningful for the QLED application especially for the commercial application. However; some questions should be addressed before accepted.

1 In Page 7, why take the 10 mA cm^{-2} for testing the CdSe/CdS, and for the CdSe/CdS/ZnS and CdSeS/ZnSeS/ZnS with 100 mA cm^{-2} ? Why use the 30 mA cm^{-2} but not the 100 mA cm^{-2} for the testing?

Our revision and responses: We thank the reviewer for the valuable comment. The purpose of measuring the time-dependent PL QY of the QDs in single-carrier devices is to investigate the redox activity of the QDs. The CdSe/CdS QDs with $\text{Cd}(\text{RCOO})_2$ ligands are more reactive than the CdSe/CdS/ZnS QDs and CdSeS/ZnSeS/ZnS QDs with $\text{Zn}(\text{RCOO})_2$ ligands, because the standard reduction potential of Zn^{2+} is $\sim 0.3 \text{ V}$ more negative than that of Cd^{2+} (Supplementary Fig. 2). Therefore, a current density of 10 mA cm^{-2} applied to the electron-only devices with the CdSe/CdS QDs is sufficient to demonstrate that the CdSe/CdS- RNH_2 QDs are stable while the CdSe/CdS- $\text{Cd}(\text{RCOO})_2$ QDs are not. In contrast, a higher current density of 100 mA cm^{-2} is applied to the electron-only devices with the CdSe/CdS/ZnS QDs (or the CdSeS/ZnSeS/ZnS QDs) to distinguish behavior of the QDs with different surface ligands. Regarding the hole-only devices (ITO/PEDOT:PSS/poly-TPD/PVK/QDs/CBP/ MoO_x /Au), the injection of electrons becomes evident when the current density is larger than $\sim 50 \text{ mA cm}^{-2}$ (corresponding to a high driving voltage of $\sim 10 \text{ V}$, see figure below). Therefore, a current density of 30 mA cm^{-2} is applied to ensure that the charge carriers transported in the hole-only devices are exclusively holes, but not electrons.

In the revised manuscript, we have added a few sentences in the captions of Fig.2 (Page 24, highlighted) and Fig. 4 (Page 26, highlighted) in the revised manuscript to explain the conditions used for the single-carrier devices.

2 In Page 8, the author demonstrated that completely replace all carboxylate and cadmium-carboxylate ligand, however the $-COO$ peak appeared in the Fig. 3a. How to explain this?

Our revision and responses: Sorry for the confusion caused by the peak labels in Fig. 3a of the original manuscript. For the sake of clarity, we have marked the peak at $\sim 1460\text{ cm}^{-1}$, which is originated from the bending vibration of $-CH_2-$ groups in either types of ligands. The revised Fig. 3a (also see below) demonstrates that the cadmium-carboxylate ligands have been completely replaced by the amine ligands.

Responses to Reviewer #3

The paper "Electrochemically-stable ligands bridge the photoluminescence..." is very interesting and well written paper. It emphasizes that the issue that material design requirements for EL applications does not always coincide with what is good enough for PL applications. The device performance is impressive and as far as the device performance and its relation to the presence of different ligands, I find the paper to be rigorous and providing several different indications that converge to the same conclusion. If you choose the ligand wrong – your film PL will have very little in common with its PL.

There is also significant part of the paper that mentions electrochemical activity. Unfortunately, I am not an expert in this. I can't tell if the improved results are due to better coverage and/or passivation of the surface or that there is indeed an electrochemistry that involves one ligand and not the other. I hope that this important issue can be answered by one of the other referees.

Assuming this part is correct than I believe the paper is suitable for publication.

Our revision and responses: We thank the reviewer for his/her encouraging comments. No action needed.

REVIEWERS' COMMENTS:

Reviewer #1 (Remarks to the Author):

this paper is ready for publication now, in my opinion.

Reviewer #2 (Remarks to the Author):

All my questions were addressed reasonable, it could be accepted.

Reviewer #3 (Remarks to the Author):

In continuation to my first report. I have reviewed the response to referee comments, and I found that all has been answered adequately.